| Editor's Pick | Genomics and Proteomics | Research Article

# Genomic comparison and phenotypic characterization of *Pseudomonas aeruginosa* isolates across environmental and diverse clinical isolation sites

Cristina Penaranda,[1,2,3,4] Evan P. Brenner,[5] Anne E. Clatworthy,[3,4] Lisa A. Cosimi,[4] Janani Ravi,[5] Deborah T. Hung[3,4,6]

**ABSTRACT**    *Pseudomonas aeruginosa* is a clinically significant, opportunistic pathogen adept at thriving in both host-associated and environmental settings. We sought to define the extent to which *P. aeruginosa* isolates specialize across niches using a comprehensive study of whole-genome sequencing with paired phenotypic characterization of 125 *P. aeruginosa* isolates from diverse clinical and environmental sites. We evaluated virulence-associated traits, including motility, cytotoxicity, biofilm formation, pyocyanin production, and antimicrobial resistance to eight antibiotics. Our results show that genomic diversity does not correlate with isolation source or most virulence phenotypes. Instead, we find that, in agreement with prior studies, the two major *P. aeruginosa* clades (groups A and B) clearly segregate by cytotoxicity, with group B strains showing significantly higher cytotoxicity than group A. Sequence analysis revealed previously uncharacterized alleles of genes encoding type III secretion effector proteins. We observed high variability among strains and isolation sources in the four assayed virulence phenotypes. Antimicrobial resistance was exclusively observed in clinical isolates, whereas it was absent in environmental isolates, reflecting antibiotic exposure-driven selection. Bacterial genome-wide association studies (GWAS) revealed an association between cytotoxicity and *exoU* presence, and we identified a novel *exoU* allelic variant with decreased cytotoxicity, demonstrating that functional diversity of well-characterized virulence factors may influence pathogenic outcomes. Overall, our analysis supports the hypothesis that the ability of *P. aeruginosa* to thrive across diverse niches is driven not by niche-specific accessory genes but by its core genome. Thus, *P. aeruginosa* isolates are capable of broad niche colonization without initial genetic adaptations.

**IMPORTANCE**  *Pseudomonas aeruginosa* is a clinically significant opportunistic pathogen adept at thriving in both host-associated and environmental niches. A major gap in our understanding of this difficult-to-treat pathogen is whether niche specialization occurs in the context of human disease. Addressing this question is critical for guiding effective infection control strategies. Previous large-scale studies have focused solely on genotypic or phenotypic analyses; when paired, they have been limited to a single phenotypic assay or to a small number of isolates from one source, or relied on PCR-based methods targeting a restricted set of genes. To comprehensively uncover niche specialization and pathogenic versatility, we performed whole-genome sequencing and phenotypic characterization of five virulence-associated traits, including antimicrobial susceptibility of 125 clinical and environmental *P. aeruginosa* isolates. Our systems-level findings challenge reductionist models of bacterial niche specialization, instead supporting an integrated view where conserved genomic systems enable opportunistic pathogenesis across diverse environments.

Address correspondence to Cristina Penaranda, penarandac@njhealth.org, Janani Ravi, janani.ravi@cuanschutz.edu, or Deborah T. Hung, hung@molbio.mgh.harvard.edu.

Cristina Penaranda and Evan P. Brenner contributed equally to this article. C.P. was responsible for wet-lab data generation, study conceptualization, and interpretation, and E.P.B. was responsible for the full data analysis and interpretation; both authors wrote the first draft of the manuscript.

Cristina Penaranda, Janani Ravi, and Deborah T. Hung are joint senior authors.

The authors declare no conflict of interest.

See the funding table on p. 14.

**KEYWORDS** *Pseudomonas aeruginosa*, genotype-phenotype correlation, antibiotic resistance, type III secretion, population structure, comparative genomics

*P*seudomonas aeruginosa is a ubiquitous environmental Gram-negative bacterium capable of both acute and chronic infections at mucosal surfaces, such as the urinary and respiratory tracts, skin, and eyes. *P. aeruginosa* is a leading cause of nosocomial infections, especially in people with cystic fibrosis (CF). Importantly, high intrinsic antimicrobial resistance (AMR) complicates *P. aeruginosa* treatment (1). Multidrug-resistant *P. aeruginosa* in the US caused an estimated 2,700 deaths in 2017 and was designated a CDC threat level of "serious" in 2019 (2).

To understand the pathogenic versatility of *P. aeruginosa*, it is essential to consider the genetic repertoire across the species. Since the first lab-reference strain, PAO1, was sequenced 25 years ago (3), over 40,000 *P. aeruginosa* genomes have been deposited to NCBI as of 2025. The pangenome of *P. aeruginosa*, composed of conserved core genes and variable accessory genes, is large and has high genetic complexity compared to other bacterial species (4). The conservation of core metabolic and virulence genes across environmental and clinical isolates is thought to explain the ability of this bacterium to survive in diverse ecological niches and host tissues (5, 6). Nevertheless, niche specialization in the context of human disease has not been defined and could be critical to effectively focus infection control efforts by targeting bacterial pathways important for the pathogenesis of a given host tissue.

Combining comparative genomic analysis with phenotypic characterization allows the correlation of specific genotypes to phenotypic differences and the identification of previously unknown pathways involved in virulence. It provides insights into the complex biology underlying pathogen evolution and diversification through the identification of conserved and novel genetic elements within a species (7). Advances in DNA sequencing technologies now permit fast and inexpensive sequencing of bacterial genomes, opening the door for methods standardly used in human population genetics, like genome-wide association studies (GWAS), to better understand bacterial genetic flow and evolution during infection (8, 9). For instance, analysis of *Campylobacter* isolates revealed that genes encoding vitamin B5 biosynthesis are associated with host specificity (10). Analysis of within-species bacterial strain variation provides insights into the exchange of genes through horizontal gene transfer and the evolution of genes through changes at the nucleotide level (7). The impacts of such changes on pathogenesis are of great interest as they may reveal the mechanisms of host–pathogen interactions and uncover novel therapeutic targets. While genomic analysis has yielded significant insights into *P. aeruginosa* population structure and diversity, bridging the gap between sequenced genotypes and resulting phenotypes remains challenging.

We report the whole-genome sequencing and comprehensive phenotypic characterization of 125 *P. aeruginosa* strains, including environmental isolates and clinical isolates that represent the major body sites that this pathogen colonizes (lung, skin, urinary tract, blood, eye). Phenotypic characterization in four virulence-associated assays, including cytotoxicity, motility, biofilm, and pyocyanin production, as well as AMR status for clinically relevant antibiotics (gentamicin, tobramycin, amikacin, imipenem, ceftazidime, piperacillin, levofloxacin, and ciprofloxacin) (Fig. 1). We did not identify distinct associations between phylogeny and isolation source, or virulence in the phenotypes assayed. Our results recapitulate the previously identified phylogenetic separation between groups A and B; our novel analysis of isolate source identified no discernible associations between phylogeny and isolation source. Moreover, there were no distinct associations between isolation source and virulence in the phenotypes assayed. We observed that AMR was frequent in clinical strains and nearly absent in environmental isolates, supporting antibiotic exposure-driven clinical AMR. Sequence analysis of the genes encoding type III effector proteins revealed previously unrecognized sequence variation, including alleles that are specific to each phylogenetic group. Our findings highlight the genetic and phenotypic diversity within the *P. aeruginosa*

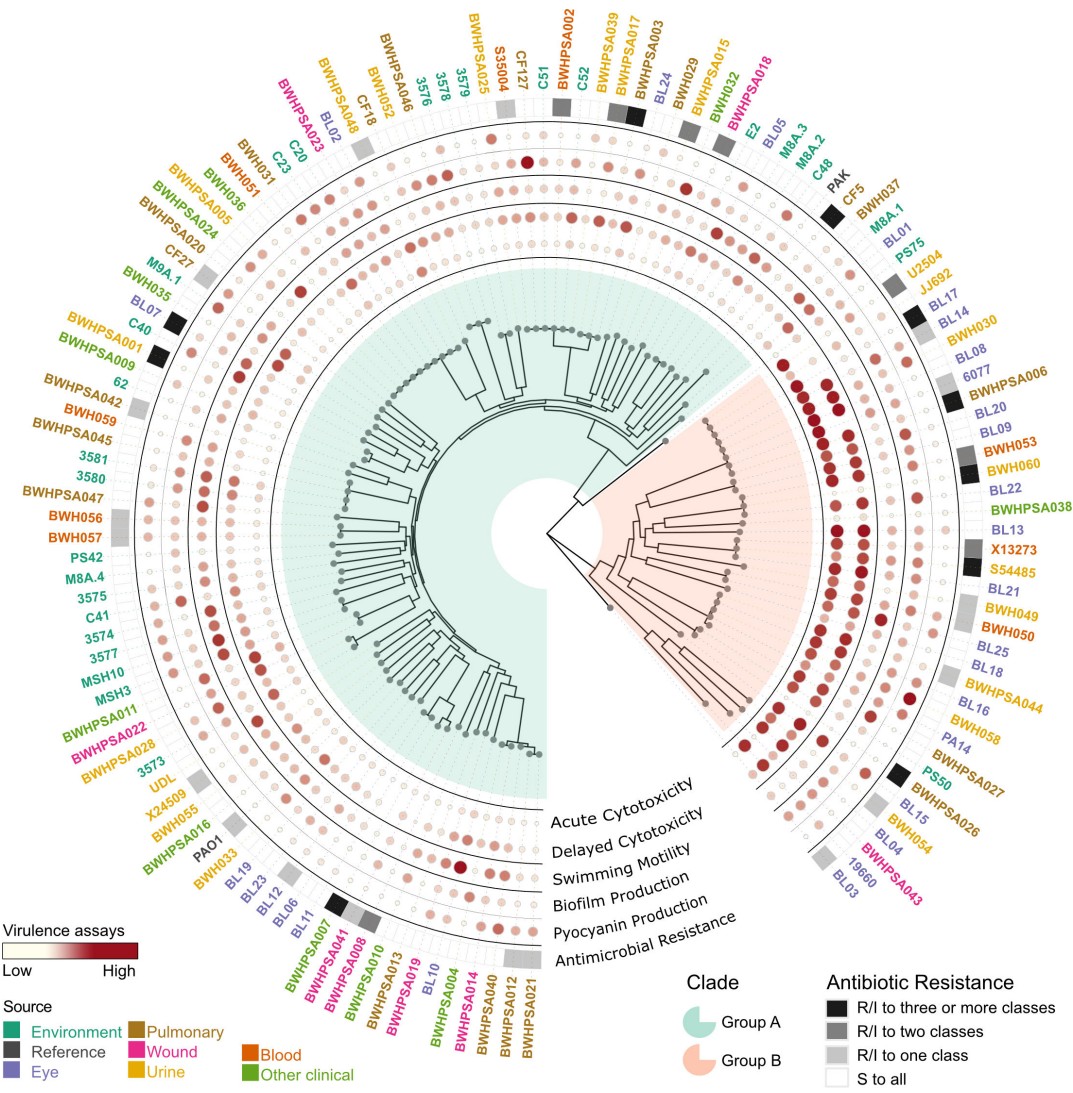

**FIG 1** Summary of phylogenetic and phenotypic data of newly sequenced *P. aeruginosa* strains. Whole-genome sequencing and phenotypic characterization of 125 clinical and environmental strains and three lab reference strains (PAO1, PAK, and PA14).

population. Collectively, these data suggest that the large core genetic repertoire of *P. aeruginosa* grants the versatility to cause infection across anatomical sites, regardless of origin.

## MATERIALS AND METHODS

### Strains

BWH/BWHPSA strains were collected by the Clinical Microbiology Laboratory at Brigham and Women's Hospital (Boston, MA). Collection of discarded, de-identified bacterial strains was approved as non-human subjects research by the Partners Healthcare Institutional Review Board (2009P001837). Eye (BL) strains were obtained from Wolfgang Haas (Bausch & Lomb). Environmental strains were obtained from Roberto Kolter, Harvard Medical School (Boston, MA), and Paula Suarez, Simon Bolivar University (Venezuela). Other strains were obtained from Frederick M. Ausubel, Massachusetts General Hospital (Boston, MA), and Steve Lory, Harvard University (Boston, MA). We aimed to represent a diversity of infections since previous analysis has focused mainly on strains isolated from the lungs of people with CF. Our collection of clinical strains

includes 17 strains previously analyzed by Wolfgang et al. by microarray (6), as well as 59 new strains collected at Brigham and Women's Hospital (Boston, MA) and 25 eye isolates provided by Bausch & Lomb. The latter set was chosen to provide geographical diversity to our collection. Importantly, the strains collected at Brigham and Women's Hospital were only passaged once before sequencing and phenotyping, with minimal opportunity for mutations to be acquired outside of the host. Our analyses also included the previously genome-sequenced laboratory strains PAO1 (3), PA14 (11), and PAK (12). Metadata, including isolation source, is presented in Data S1.

## Whole-genome sequencing

Strains were grown overnight in lysogeny broth (LB). Genomic DNA was purified using the DNeasy Blood & Tissue kit (Qiagen) per the manufacturer's protocol. Whole-genome sequencing was performed at the Broad Institute Genomic Core using paired-end sequencing. Libraries were sequenced on the Illumina HiSeq 2000 platform and sequencing runs deposited in the Sequence Read Archive (SRA) (Data S1).

## Genome assembly and analysis

For assembly, paired-end reads for all 125 successfully sequenced isolates were downloaded from NCBI SRA using SRA Toolkit (13) (v3.0.0) to the University of Colorado Alpine HPC cluster. Reads were processed by FASTP (14) (v1.0.1) for QC and adapter trimming, then assembled *de novo* with Shovill (15) (v1.1.0). Genome annotation was completed for all samples by Bakta (16) (v1.9.4) using database version 5.1 and a Prodigal training file using *P. aeruginosa* PAO1. Genome assembly and annotation completeness were assessed through QUAST (17) (v5.3.0) and BUSCO (18) (v5.8.3) with BUSCO data set odb12 (19), respectively.

To explore variation across our isolates plus three reference genomes, PAO1, PA14, and PAK, Panaroo (20) (v1.5.2) was used to generate a 128-strain pangenome. After processing, pseudogenes were removed for downstream analysis. Scoary2 (21) (v0.0.15) was used for measuring statistical associations between gene presence and phenotypic traits, using the pangenome gene presence/absence matrix, and a phylogenetic tree that was produced by IQ-TREE2 as described below. Finer variation was extracted using Snippy (22) (v4.6.0) and the PAO1 reference to extract polymorphisms across all isolates. For analysis within phylogenetically distinct group B isolates, Snippy was run a second time against the group B PA14 reference to explore polymorphisms in group B-specific genes like *exoU*.

## Phylogenetic analysis

All called single-nucleotide polymorphisms (SNPs) against PAO1 were used as the core SNP set to infer a phylogenetic tree with IQ-TREE2 (23) (v2.4.0), with the best-performing substitution rate model, GTR+F+I+R6, chosen through ModelFinder (24) (v0.1.7) (Data S2). The resulting phylogenetic tree was tested using 1000 SH-like aLRT replicates (25, 26) implemented in IQ-TREE (Data S2 and S3). The phylogenetic tree was visualized and figures generated through R (v4.4.1) in RStudio (v2025.05.1), and R packages BiocManager (27) (v3.19), ggplot2 (28) (v3.5.2), ggtree (29) (v3.12.0), ggtreeExtra (30) (v1.14.0), ggnewscale (31) (v0.5.1), ape (32) (v5.8.1), tidyr (33) (v1.3.1), dplyr (34) (v1.1.4), RColorBrewer (v1.1.3), ggsignif (35) (v.0.6.4), and svglite (36) (v2.2.1).

## Resistance gene analysis

Presence of known resistance-associated genes was identified in assemblies using AMRFinderPlus (37) (v4.0.3) with database 2025-06-03.1.

## Swimming assay

Swimming motility was measured on 0.35% agar plates as previously described with minor modifications (38). Briefly, a sterile toothpick was used to pick individual colonies

grown on LB plates and stabbed into the agar layer of a fresh 0.35% agar plate. Plates were incubated upright for 16–17 h at 30°C. Images were acquired on a gel imaging station. The diameter of the swimming zone was measured using ImageJ (39). Each strain was assayed at least twice, and the average swimming zone was calculated. Phenotypic results are available in Data S1 for all assays.

## Biofilm assay

Biofilm formation was measured using crystal violet as previously described (40). Log-phase bacteria grown in LB ($OD_{600}$ = ~0.3) were diluted to $2.5 \times 10^6$ CFU/mL and plated in tissue culture-treated 96-well plates at 100 µL/well as eight replicates. Plates were incubated for 24 h at 30°C. Plates were washed in water and stained with 200 µL/well 0.1% crystal violet, incubated for 30 min at RT, and washed three more times in water. A volume of 100 µL/well acetic acid (33%) was added, and absorbance was measured at 590 nm. Strains were assayed two or three times. All technical and biological replicates were averaged.

## Pyocyanin assay

Pyocyanin was extracted from the supernatant fraction of strains grown overnight in LB as previously described (41). All strains were assayed in triplicate.

## Epithelial cell cytotoxicity assay

Host cell viability was measured using CellTiterGlo at 4 h (acute cytotoxicity) and 24 h (delayed cytotoxicity) post-infection in a gentamicin protection assay, as previously described (42). Briefly, 5,637 human bladder epithelial cells (RRID: CVCL_0126) were seeded overnight in RPMI media supplemented with 10% FBS. Log-phase bacteria grown in LB ($OD_{600}$ = ~0.3) and diluted in media were used to infect cells at MOI = 1 in triplicate and centrifuged to synchronize infection. Cells were incubated for 2 h, extracellular bacteria were removed, media containing 200 µg/mL gentamicin was added, and incubated for an additional 2 h, and cell viability was measured at 4 h. For measurement of delayed cytotoxicity, the media were replaced with media containing 25 µg/mL gentamicin to prevent reinfection, and cell viability was measured at 24 h. The assay was repeated at least three times for most of the strains. Some strains could not be assayed for delayed cytotoxicity due to gentamicin resistance (shown as "n/a").

## Antibiotic susceptibility assay

Antibiotic susceptibility was assayed using BBL Sensi-Discs (BD Diagnostics): ciprofloxacin (5 µg), ceftazidime (30 µg), imipenem (10 µg), piperacillin (100 µg), levofloxacin (5 µg), tobramycin (10 µg), gentamicin (10 µg), and amikacin (30 µg). Muller-Hinton agar plates were inoculated with overnight bacterial cultures grown in LB. Plates had 4–5 discs placed and were incubated for 16 h at 37°C. Images were acquired on a gel imaging station, and the diameter of the zone of inhibition was measured using ImageJ. Most strains were assayed at least twice per antibiotic, and the average zone of inhibition was calculated. Susceptibility was determined based on the manufacturer's Zone Diameter Interpretive Chart for *P. aeruginosa* reference strain ATCC 27853.

## *exoU* allele cloning and overexpression

Alleles of *exoU* were amplified from PA14 and BL18 (using 5′ TTCGGTACCATGCATATCC AATCGTTGG and 5′ ATTAAGCTTTCAAACGAACACTAACGC primers) and cloned into the pHERD20T vector using KpnI and HindIII. Plasmids were transformed into a PA14Δ*exoU* strain.

## RESULTS

### Phylogenetic diversity

Advances in bacterial genomic sequencing technologies enable rapid and inexpensive means to identify genetic features that lead to adaptation in the environment and during infection (8, 9). To gain insights into the pathogenesis of *P. aeruginosa* and, specifically, niche specialization in the context of human disease, we assembled a diverse collection of clinical isolates (*n* = 97) from blood (*n* = 9), eye (*n* = 27), pulmonary (*n* = 20 for people with [*n* = 4] and without [*n* = 16] CF), urine (*n* = 22), wound (*n* = 8), and other infections including ear infections and abscesses (*n* = 11), as well as environmental isolates (*n* = 28) (Fig. S1). After paired-end Illumina whole-genome sequencing, we performed *de novo* genome assembly on all 125 strains. The number of contigs for the final assemblies ranged from 76 to 425. The average genome size was 6.65 Mbp, with an average CDS count of 6,074 (Fig. S1). We constructed phylogenetic trees that included the reference strains PAO1, PA14, and PAK based on the entire concatenated SNP set extracted across all genomes. This set of 128 strains, comprising the 125 newly sequenced strains and the three reference strains, was used for all subsequent analyses.

Recapitulating prior analyses, the population structure consists of two major clades (7, 43): 92 strains, including the reference strains PAO1 and PAK, belong to group A, while 34 strains, including PA14, belong to group B (Fig. 2A; Fig. S2A). Two strains (PS75 and BL03) group separately from the major clades, potentially belonging to minor clades, as has been previously suggested (43). None were phylogenetically related to the taxonomic outlier PA7 (Fig. S2A) (44). Analysis by Ozer et al. of intra- versus inter-group recombination events suggested that groups A and B inhabit distinct ecological niches (43); our results only support this conclusion for environmental strains, which are overrepresented among group A isolates (Fig. S2B and C). However, all other clinical strains were evenly divided between groups A and B.

Strains from the same anatomic site of infection did not cluster on the phylogenetic tree (Fig. 2A), in line with previous population-wide analysis (6). We also analyzed the accessory genome by examining the gene presence/absence patterns per genome by t-SNE to determine whether strain-specific accessory genes correlated with anatomic site of infection; we did not see evidence of isolation source correlating with pangenomic accessory gene content (Fig. 2B). No statistically associated genes were found between isolation sources (Data S4).

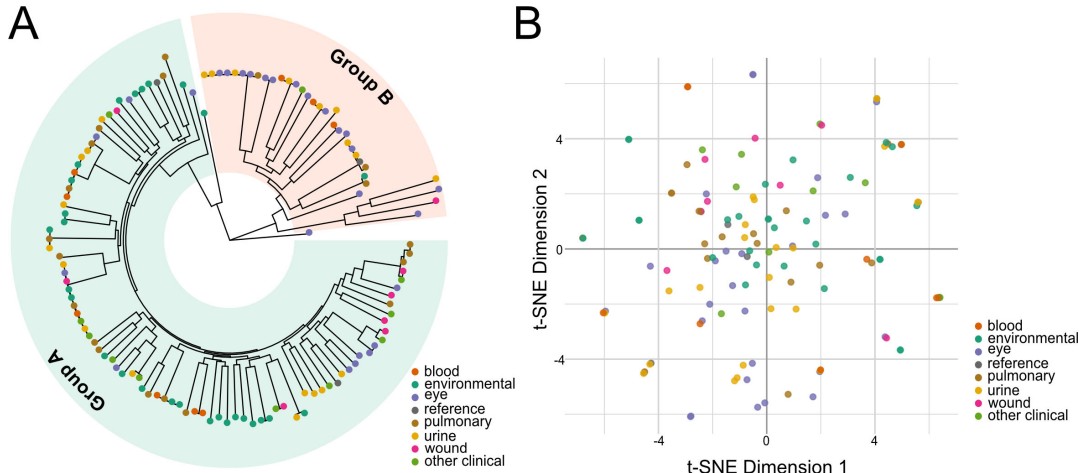

**FIG 2** Phylogenetic distribution of newly sequenced *P. aeruginosa* strains. (A) Circular phylogenetic tree based on nucleotide positions with sequencing coverage across all strains. Strains are colored at the tree tips by isolation source, and broader phylogenetic groups are highlighted. (B) t-SNE representation of the accessory genome. Genes present in <95% of strains were considered accessory and used for this analysis. The isolation source had little correlation with phylogeny and accessory genome content.

## Sequence analysis of type III effectors

*P. aeruginosa* encodes four type III secretion system (T3SS) effector proteins (ExoU, ExoT, ExoS, and ExoY) that target the eukaryotic membrane and cytoskeleton and cause cytotoxicity (45). The prevalence of the four effector genes *exoY*, *exoT*, *exoS*, and *exoU* in clinical and environmental strains has been widely studied due to their critical role in disease (7, 46, 47). In agreement with previous reports (46), we found that the *exoS* or *exoU* genes are mutually exclusive: 93 strains carry intact *exoS*, while the remaining 33 strains carry *exoU* (Fig. 3). Furthermore, all 128 strains carry the *exoT* gene, supporting that this is not a variable trait (43). We identified a frameshift mutation in CF5 (*exoT* I182fs) that likely results in a non-functional protein (Data S5). Therefore, although all strains carry the *exoT* gene, not all strains appear to have a functional ExoT protein.

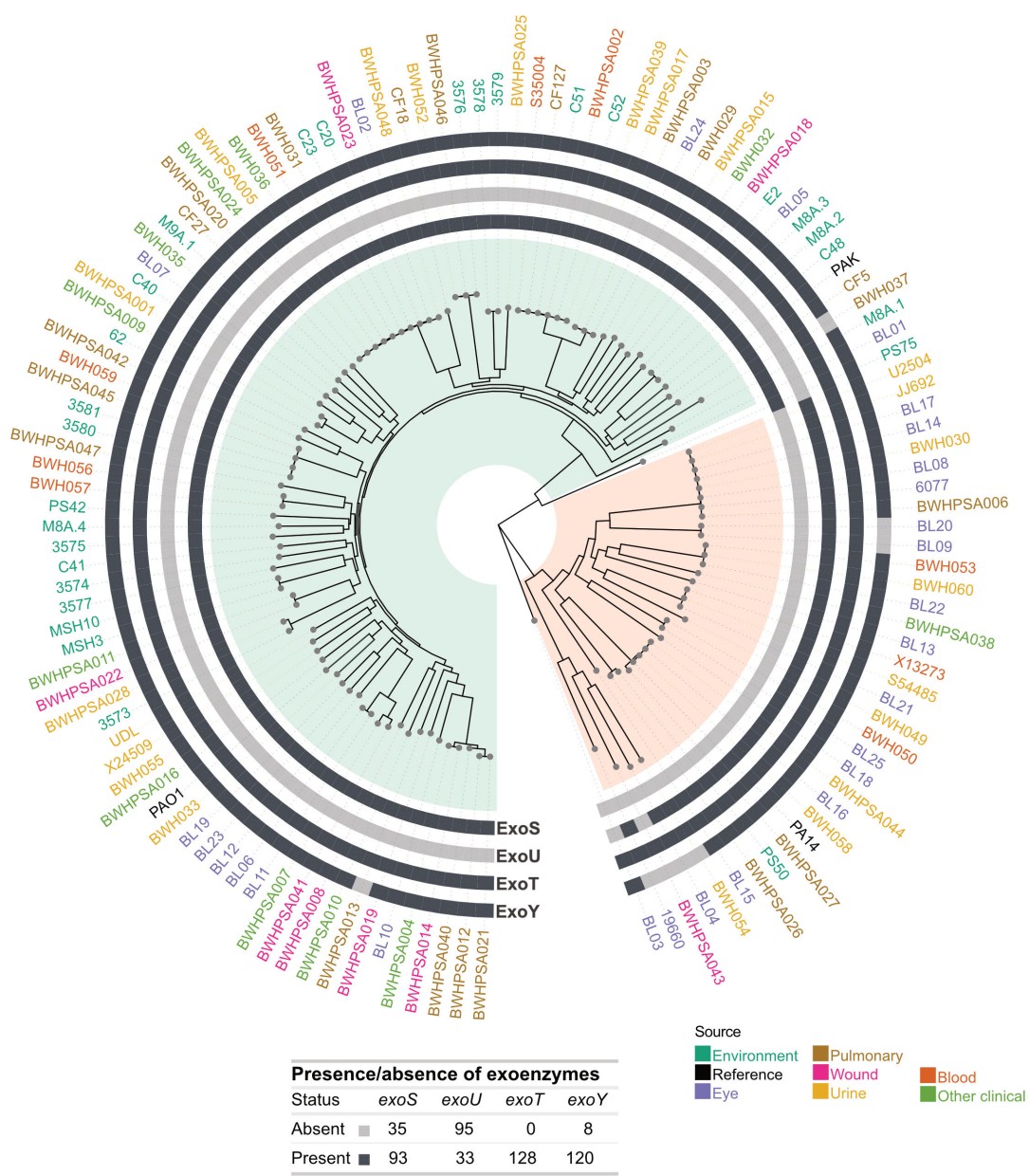

| Presence/absence of exoenzymes | | | | |
|---|---|---|---|---|
| Status | *exoS* | *exoU* | *exoT* | *exoY* |
| Absent | 35 | 95 | 0 | 8 |
| Present | 93 | 33 | 128 | 120 |

**Source**
- Environment
- Reference
- Eye
- Pulmonary
- Wound
- Urine
- Blood
- Other clinical

**FIG 3** Genomic analysis of type III effectors. Circular phylogenetic tree showing the presence or absence of the four type III effectors (*exoY*, *exoU*, *exoT*, *exoY*) in the outer rings for each strain. Major groups A and B are highlighted in green and orange, respectively. The presence or absence of type III effectors across strains is tabulated. Strain names are colored by isolation source, as in Fig. 1.

Comparative analysis of all effector gene sequences revealed wide variation within the population (Data S5). As has been previously described, the PA14 version of ExoY has an altered C terminus compared to PAO1, resulting from a frameshift mutation (F374fs) that adds 36 amino acids (48, 49). Of the 121 strains that carry an *exoY* gene, eight strains encoded either a 410aa ($n = 3$) or 414aa ($n = 5$, including PA14) protein, while 93 (including PAO1) strains encoded the PAO1-like 378aa protein. All strains carrying a PA14-like *exoY* sequence belong to group B, suggesting that this frameshift mutation arose after the two phylogenetic groups diverged. We also identified five additional frameshift or premature stop mutations that are predicted to result in proteins of different lengths, which have also been previously reported (50). Strains C51 (Tyr186*), 3575 (Asn12fs), and BWHPSA015 (Trp359fs) each showed group A frameshifts resulting in protein lengths unique to that strain. Finally, Leu243Fs was seen in 10 group B isolates, yielding a predicted 248aa variant not annotated as ExoY by Bakta, but clustered with ExoY sequences by CD-HIT in Panaroo.

Additionally, we observed a novel polymorphism in the C terminus of the *exoT* gene that is predicted to encode a protein of different length (Data S5). The PAO1 sequence encodes a 457aa-long protein and is shared by 120 strains, while only four strains share the PA14 SNP (Gln443*), resulting in a shorter 442aa-long protein. There was a frameshift mutation in one strain (CF5 G181fs) that is also predicted to result in a shorter protein in group A.

For *exoU* or *exoY*, we did not observe alleles predicted to result in proteins of different lengths.

## Phenotypic diversity

To understand whether virulence phenotypes or AMR correlate with isolation source and phylogeny, we characterized all strains with four functional *in vitro* virulence assays: (i) production of pyocyanin, a phenazine compound that induces eukaryotic cell oxidative stress (51), (ii) flagellum-dependent swimming motility (38), (iii) formation of biofilms, associated with nosocomial infections (52), and (iv) acute (4 h) and delayed (24 h) host cell cytotoxicity, primarily attributable to injection of effector proteins into the host cytosol by the T3SS (53). Additionally, we characterized AMR to eight antibiotics spanning five major classes: quinolones (ciprofloxacin, levofloxacin), aminoglycosides (tobramycin, gentamicin, amikacin), cephalosporins (ceftazidime), carbapenems (imipenem), and ß-lactamase inhibitors (piperacillin).

There was high variability amongst strains and isolation sources in the four virulence phenotypes assayed (Fig. 4A through E). Notably, environmental strains were significantly more motile than pulmonary or clinical strains from other sources, which showed the least motility. Eye strains demonstrated significantly lower host cell survival (increased cytotoxicity) and increased biofilm production than pulmonary and environmental strains. Pulmonary strains included both CF ($n = 4$) and non-CF ($n = 16$) isolates; however, these low sample numbers did not provide sufficient statistical power for genotype-phenotype analyses.

Biofilm formation and motility have been previously reported to be positively correlated, and it has been hypothesized that movement toward and attachment to a surface are required for biofilm formation (54, 55). However, we did not observe a correlation between biofilm formation and swimming motility either for the entire population or when analyzed for each isolation site individually (Fig. 3; Data S6). It is nevertheless possible that other types of motility, such as twitching or swarming motility, are required for biofilm formation.

Within our strain collection, antibiotic resistance was widespread for clinical isolates but absent for environmental isolates (Fig. 5; Fig. S4). Resistance to at least one antibiotic was observed in 36 of 97 clinical strains, 18 strains were resistant to at least two drug classes, and 10 showed clinical multidrug resistance (defined by resistance/intermediate resistance to three or more antibiotic classes), highlighting the overall high level of AMR seen in the population (Fig. 5B). Resistant strains did not cluster phylogenetically (Fig.

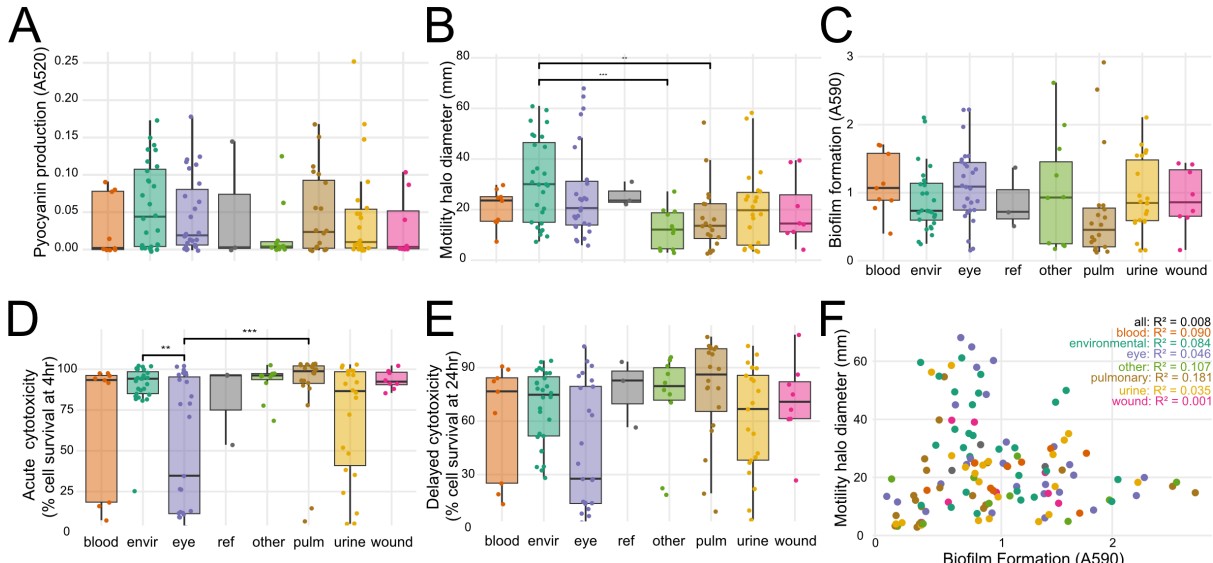

**FIG 4** Phenotypic diversity of newly sequenced *P. aeruginosa* strains. (A) Pyocyanin production in the supernatant fraction of overnight LB cultures. (B) Swimming motility on 0.35% agar plates. (C) Biofilm formation measured by crystal violet staining. (D) Acute cytotoxicity of human bladder epithelial cells at 4 h post-infection. (E) Delayed cytotoxicity of human bladder epithelial cells at 24 h post-infection. (F) Correlation of swimming motility and biofilm formation per strain. Spearman $R^2$ for each isolation group is shown. For panels A–E, Wilcoxon test with Benjamini-Hochberg correction for multiple testing: **$P < 0.01$; ***$P < 0.001$.

S4). None of the 28 environmental strains showed any resistance. This aligns with prior observations (56) and supports the idea that the high incidence of antibiotic resistance among clinical strains is likely due to selection pressure from treatment of *P. aeruginosa* infections.

We were able to identify specific genetic determinants underlying antibiotic resistance phenotypes for most resistant strains using AMRFinderPlus (Data S7). Amikacin and gentamicin resistance in BWHPSA008, and levofloxacin resistance in BWHPSA044 and BWH050 did not have distinct resistance-conferring mechanisms. For these exceptions, AMRFinderPlus identified genes generally linked to resistance, but these genes were also identified in susceptible strains for the same drugs, suggesting they may not be sufficient to confer the observed phenotype (Data S7). It is possible that changes in gene expression or protein activity are responsible for the resistance observed.

Finally, we also found that there was no significant positive correlation between AMR and the virulence phenotypes of the clinical isolates, in line with previous reports (57).

## Genotype-phenotype correlations

To identify genotype-phenotype correlations, we performed microbial GWAS for each of the four virulence phenotypes using Scoary2, our pangenomic presence/absence gene matrix, and our phenotypic data. Cytotoxicity (4 and 24 h) showed a statistically significant association with the presence of *exoU* (Data S4), which has been documented to be the primary contributor to mammalian cell cytotoxicity (45, 53, 58, 59). Correlation of swimming motility, biofilm formation, and pyocyanin production to gene content did not yield any significant gene associations, likely because of the genetic complexity of these mechanisms (Data S3). More generally, comparison of our phenotypic data to our phylogenetic data showed that only cytotoxicity and biofilm formation are segregated based on phylogeny (Fig. S3A through C).

We noted that while most strains in group B showed high acute cytotoxicity, as evidenced by low host cell survival, six strains showed greater than 70% host cell survival at this time point (Fig. 6). The lack of cytotoxicity was still evident at 24 h, demonstrating

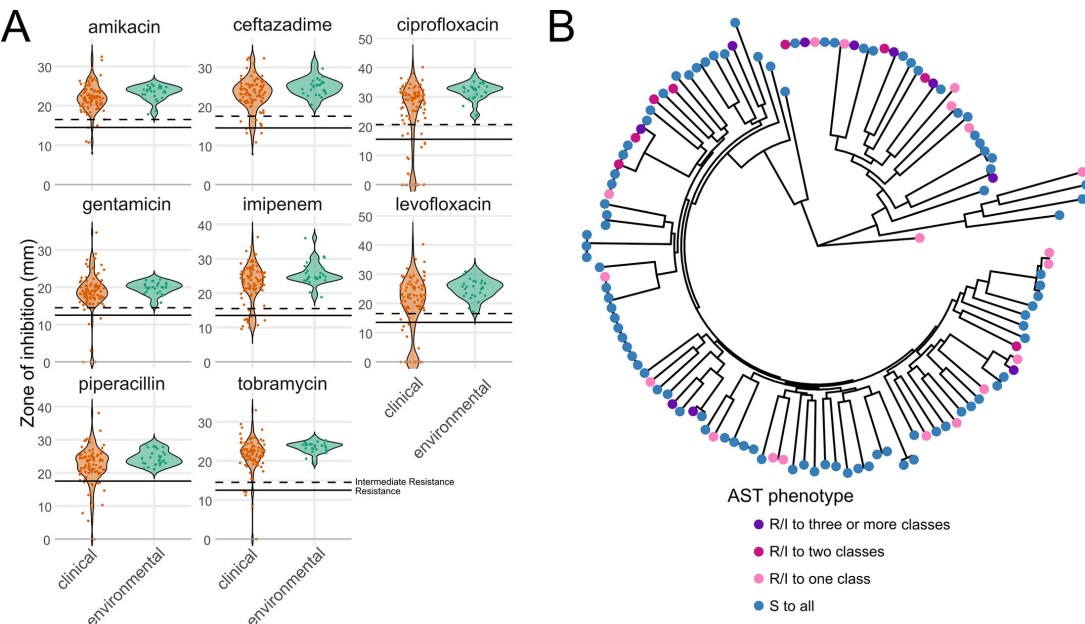

**FIG 5** Antibiotic susceptibility of *P. aeruginosa* strains. (A) Zone of inhibition for each strain to eight antibiotics representing five classes commonly used to treat *P. aeruginosa* infections: quinolones (ciprofloxacin, levofloxacin), aminoglycosides (tobramycin, gentamicin, amikacin), cephalosporins (ceftazidime), carbapenems (imipenem), and β-lactamase inhibitors (piperacillin). Solid lines indicate resistance, and dashed lines indicate intermediate resistance as based on the Zone Diameter Interpretive Chart for *P. aeruginosa* reference strain ATCC 27853. Clinical strains are shown in orange; environmental strains are shown in green. (B) Circular phylogenetic tree as in Fig. 1A showing AMR phenotype.

that this phenotype was not time-dependent (Fig. 7). Analysis of the syntenic region encoding *exoU* demonstrated genetic rearrangement and deletion of *exoU* in BWH043; no other genes were found elsewhere in the genome with >70% homology to *exoU*, suggesting that this strain is not cytotoxic because it does not carry the *exoU* gene (Fig. 6). BWHPSA026 showed no evidence of acute cytotoxicity (4 h survival mean > 100%) (Fig. 7), although its nearest phylogenetic neighbor, PS50, shows typical cytotoxicity for group B (4 h survival mean = ~25%). These strains encode identical *exoU* genes and differ by only 132 missense variants in protein-coding genes (Fig. S5A). The missense mutations present in BWHPSA026 and absent in PS50 do not seem to explain the difference in cytotoxicity (Data S8).

Sequence analysis showed that the *exoU* allele of JJ692 and BL18 had SNPs compared to the PA14 *exoU* allele (Data S5). The JJ692 SNP (Pro447Leu) is also found in other strains that were cytotoxic and therefore does not explain the decreased cytotoxicity (Data S5). Overexpression of the BL18 *exoU* allele in the PA14Δ*exoU* background resulted in lower cytotoxicity compared to overexpression of the WT PA14 *exoU* allele, suggesting that these point mutations may affect the ExoU phospholipase activity and, in isolation, result in lower host cell death (Fig. S5B). These five BL18 SNPs have not been previously described and cluster around the active site residue (Asp344) required for lipase activity (59), suggesting that they may affect protein function (Fig. S5C).

The remaining two strains (BWHPSA038 and BWH060) encoded WT copies of *exoU*. We noted that these strains were also defective in the other acute virulence phenotypes tested (pyocyanin production, swimming motility, and biofilm production, Fig. 7), which suggested that they may carry a mutation in a master regulator controlling all these phenotypes. In fact, we found unique mutations, not found in other strains in our collection, that could be responsible for a loss of all virulence phenotypes: BWHPSA038 has a frameshift mutation (Ala144fs) in the anti-sigma factor MucA resulting in a truncated protein that lacks 51 amino acids at the C-terminus, and BWH060 has a premature stop codon (Gln43*) for the type III secretion transcriptional activator ExsA (Fig. S5D). The *mucA* frameshift mutation has been previously identified in mucoid

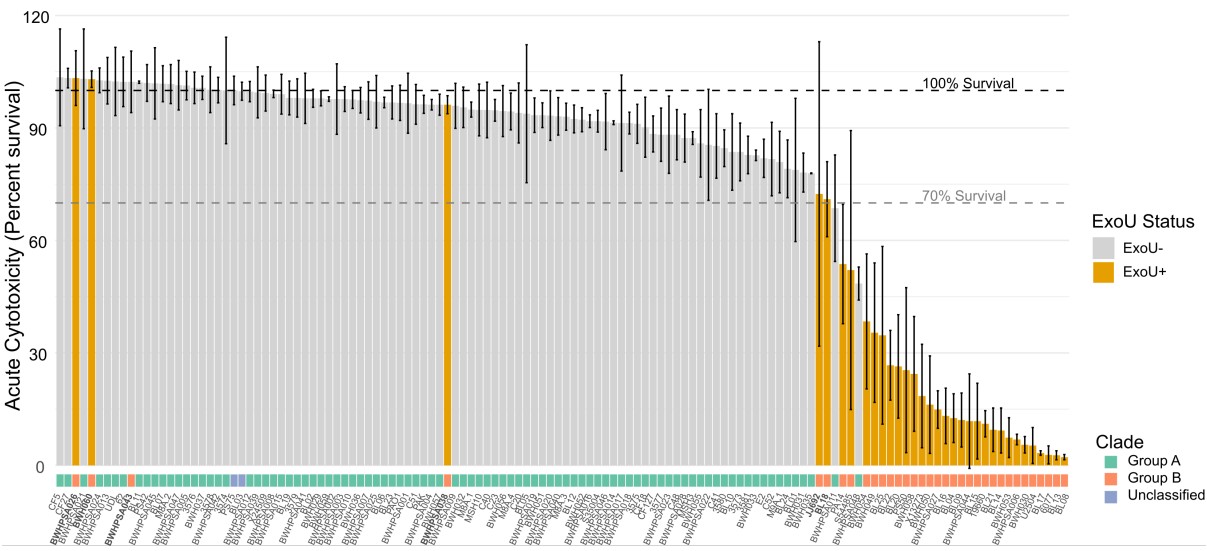

**FIG 6** Association of cytotoxicity with *exoU* and clade. Acute cytotoxicity is shown as percent survival of host cells per strain in the absence (gray) or presence (orange) of *exoU*. Clade designation is shown below. Bolded strains are those that belong to group B but show >70% mean host cell survival.

strains as the *mucA22* allele, and a similar deletion (*mucAΔG440*) was shown to result in repression of the T3SS (60, 61). Analysis of the functional domains of ExsA has demonstrated that the C-terminal domain, missing in the BWH060 strain, is necessary and sufficient for activation of type III secretion promoters (61).

## DISCUSSION

Comparative bacterial genomics to correlate genomic content with niche specialization, virulence, or AMR can reveal novel mechanisms that drive pathogenic processes, especially for diverse species like *P. aeruginosa*. Here, we performed whole-genome sequencing of 125 clinical and environmental isolates, phenotyped them with four virulence assays, and determined AMR phenotypes to eight antibiotics, generating a unique data set that integrates comprehensive genomic data with multiple phenotypic assays. The high variability of phenotypes amongst clinical and environmental strains

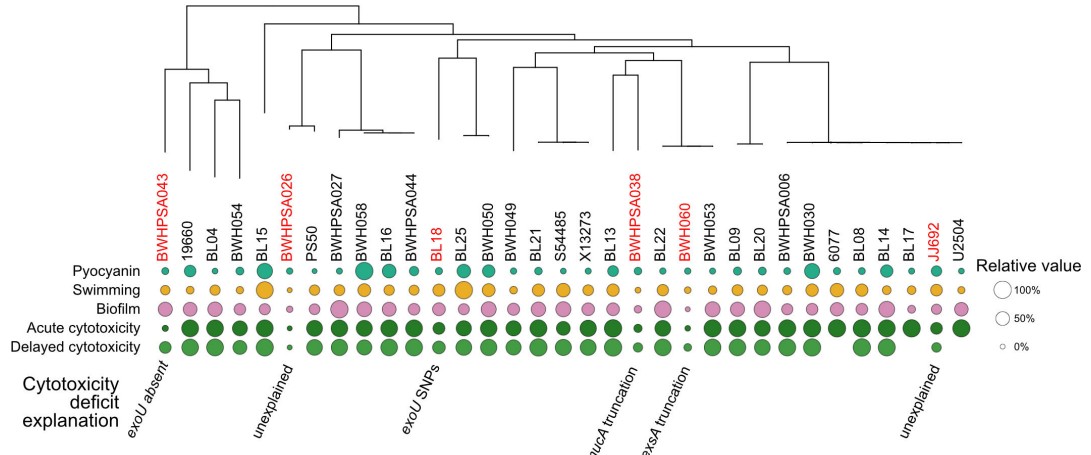

**FIG 7** Genetic mutations of strains in phylogenetic group B likely explain the non-cytotoxic phenotype. Six group B strains (red) showed more than 70% acute host cell survival in the cytotoxicity assay. This attenuated cytotoxicity phenotype is still evident at 24 h. We were able to provide potential explanations for the reduced cytotoxicity in four of these strains. For strains BWHPSA038 and BWH060, a deficit was seen across all tested virulence assays, which led to the identification of SNPs in the master regulators MucA and ExsA, respectively. Missing 24 h cytotoxicity values for strains U2504, BL17, and 6077 indicate complete host cell killing before this time point.

demonstrates the diversity within the population and that isolates are capable of broad niche colonization without initial genetic adaptations. Moreover, the low level of correlation between isolation source and phenotype suggests that none of these *in vitro* phenotypes are unconditionally required for infection of a particular host tissue.

Despite varied virulence phenotypes across the population, cytotoxicity was the only phenotype with a statistically and biologically meaningful gene by bacterial GWAS. The presence of *exoU* drove high levels of acute cytotoxicity; the role of the T3SS effector ExoU in cytotoxicity has been well-documented (46, 47, 53, 59). Nevertheless, our analysis did uncover a novel *exoU* variant that we demonstrated yields reduced cytotoxicity. Correlation of other phenotypes to gene content did not reveal novel pathways, likely because those phenotypes are multifactorial, controlled by many mechanisms. Our results suggest that, given the genetic diversity within the *P. aeruginosa* population, such analysis will likely require thousands, rather than hundreds, of strains.

We analyzed accessory genome content across isolation sources in our collection to identify genetic loci that may confer a predisposition to infection in a particular body site. However, we did not identify any genomic regions that uniquely correlated with the isolation source. Similarly, the virulence phenotypes we measured generally did not correlate with isolation sources. Limited exceptions were seen: environmental isolates showed significantly lower cytotoxicity and greater motility than a subset of clinical isolates, and AMR was absent in environmental isolates and present in many clinical isolates. A previous report suggested that host adaptation results in genome reduction (62); however, we did not observe a difference in genome size or CDS content between environmental and clinical strains. It is important to note that the clinical data associated with the clinical strains in our collection do not differentiate between long-term and short-term infections. Therefore, it remains possible that long-term adaptation to a mammalian host may result in genome size reduction due to interplay with host factors during persistence, as has been observed in people with cystic fibrosis (63). Altogether, our results indicate that disease-causing strains are not distinctly genetically dissimilar from those found in the environment.

Our analysis also allowed us to correlate virulence phenotypes and antibiotic susceptibility. Biofilm formation and motility have been previously correlated, and it has been hypothesized that movement toward and attachment to a surface are required for biofilm formation (54, 55). However, we did not see a positive correlation between biofilm formation and flagella-dependent swimming motility. This finding is in line with a recent analysis using an independent collection of strains that also did not detect a correlation between biofilm formation and any form of motility (swimming, swarming, twitching) (57).

T3SS has a critical role in pathogenesis as it is the major contributor to mammalian cell cytotoxicity, enabling bacteria to directly inject effector proteins into the cytoplasm of host cells. The mechanisms by which ExoU, ExoT, ExoS, and ExoY cause cell death have been widely studied and shown to target both the membrane and the cytoskeleton. PAO1 and PA14 are laboratory reference strains and have been used to examine the role of these effector proteins in various settings (64, 65). Both strains carry the *exoT* and *exoY* genes, while only PAO1 carries the *exoS* gene and only PA14 carries the *exoU* gene. As had been previously reported, the C-terminus of ExoY in PA14 is longer than in PAO1 (49). This longer allele was only found in four additional strains within our collection. We also noted that the PAO1 allele of *exoT*, which encodes a shorter protein than the PA14 allele, is much more common within our collection. This suggests that the PAO1 alleles of *exoY* and *exoT*, rather than the PA14 alleles, should be considered the WT versions and used as a reference when studying the mechanisms of action of these effector proteins.

Given the rise in AMR, identifying the mechanisms contributing to resistance has been a significant goal in recent decades. The ability of next-generation sequencing to identify novel genes based on sequence similarity has revolutionized our knowledge of resistance elements and their diversity (66). However, classification of AMR by genome sequence analysis alone remains imperfect. Numerous factors beyond the

presence/absence of a gene within a genome could culminate in AMR, including SNPs, changes in gene expression, and protein activity. As a functional readout, we must still rely on laboratory-based susceptibility testing, which is both costly and time-consuming since it must be done for each antibiotic individually. Prior genomic analysis of 390 *P. aeruginosa* strains demonstrated that susceptibility to meropenem and levofloxacin could be readily determined from sequence analysis, but determining susceptibility to amikacin was less tractable (67). In our study, we were able to assign genetic determinants of resistance in ~93% of cases where intermediate resistance or resistance was identified, demonstrating a high probability of sequence-based susceptibility classification for the antibiotics that we analyzed. As more strains are sequenced and more mechanisms of resistance are identified, sequence-based susceptibility classification should become more feasible and reliable, but particularly for *P. aeruginosa*, the regulatory networks that modulate gene expression are more complex than mere presence/absence binaries. A gene may be encoded but not sufficiently expressed to confer phenotypic resistance, which requires ASTs or other methods to discern clinical AMR status. Lastly, a more detailed understanding of mechanisms of resistance may point to novel ways to subvert these pathways, potentially allowing us to continue to rely on current antibiotics.

In conclusion, our comprehensive genomic and phenotypic analysis of 125 clinical and environmental *P. aeruginosa* isolates reveals that this opportunistic pathogen's remarkable versatility stems from its conserved genomic repertoire rather than niche-specific gene acquisition. The lack of strong genotype-phenotype correlations for most virulence traits and the absence of isolation source-specific genomic signatures highlight *P. aeruginosa*'s capacity for opportunistic colonization across diverse environments and host tissues. This finding also alludes to likely roles of other functional layers (e.g., gene expression, metabolic landscape) contributing to *P. aeruginosa*'s versatility. The apparent dichotomy between AMR patterns in clinical versus environmental strains underscores the critical role of antibiotic selection pressure in driving resistance evolution. These collective findings bolster our understanding of *P. aeruginosa* pathogenesis and evolution, and the rich genotype-phenotype data set generated here provides a valuable resource for future studies investigating the pathogenic potential of this versatile species.

## ACKNOWLEDGMENTS

We thank members of the Hung lab for helpful discussion and Roberto Kolter, Paula Suarez, Wolfgang Haas (Bausch & Lomb), Fred Ausubel, and Steve Lory for sharing their strain collections. Strains with BWH or BWHPSA nomenclature are from the Crimson Biomaterials Collection Core Facility at Mass General Brigham, Boston, MA. We are also grateful to Emily Meyer and Jerome McKay (Research Development, Department of Biomedical Informatics, University of Colorado Anschutz) for feedback on the revised manuscript.

This work was funded by grants to D.T.H. (1R21AI097613-01) and J.R. (1U01AI176414) and institutional funds to C.P. (National Jewish Health) and J.R. (institutional funds and Translational Research Scholars Program from the University of Colorado Anschutz). The funders had no role in study design, data collection, and interpretation, or the decision to submit the work for publication.

## AUTHOR AFFILIATIONS

[1]Department of Immunology and Genomic Medicine, National Jewish Health, Denver, Colorado, USA
[2]Department of Immunology and Microbiology, University of Colorado Anschutz, Aurora, Colorado, USA
[3]Department of Molecular Biology and Center for Computational and Integrative Biology, Massachusetts General Hospital, Boston, Massachusetts, USA

[4]Infectious Disease and Microbiome Program, Broad Institute of Harvard and MIT, Cambridge, Massachusetts, USA

[5]Department of Biomedical Informatics, University of Colorado Anschutz, Aurora, Colorado, USA

[6]Department of Genetics, Harvard Medical School, Boston, Massachusetts, USA

## AUTHOR ORCIDs

Cristina Penaranda (iD) http://orcid.org/0000-0002-2595-2863
Evan P. Brenner (iD) http://orcid.org/0009-0000-7067-8886
Janani Ravi (iD) http://orcid.org/0000-0001-7443-925X
Deborah T. Hung (iD) http://orcid.org/0000-0003-4262-0673

## FUNDING

| Funder | Grant(s) | Author(s) |
|---|---|---|
| National Institutes of Health | 1R21AI097613-01 | Deborah T. Hung |
| National Institutes of Health | 1U01AI176414 | Janani Ravi |
| National Jewish Health | | Cristina Penaranda |
| University of Colorado Anschutz | | Janani Ravi |

## AUTHOR CONTRIBUTIONS

Cristina Penaranda, Conceptualization, Data curation, Formal analysis, Funding acquisition, Investigation, Methodology, Project administration, Resources, Supervision, Validation, Visualization, Writing – original draft, Writing – review and editing | Evan P. Brenner, Data curation, Formal analysis, Investigation, Methodology, Software, Validation, Visualization, Writing – original draft, Writing – review and editing | Anne E. Clatworthy, Conceptualization, Methodology, Resources, Supervision, Writing – review and editing | Lisa A. Cosimi, Resources, Writing – review and editing | Janani Ravi, Formal analysis, Funding acquisition, Investigation, Methodology, Project administration, Supervision, Visualization, Writing – original draft, Writing – review and editing | Deborah T. Hung, Conceptualization, Funding acquisition, Investigation, Methodology, Project administration, Resources, Supervision, Visualization, Writing – review and editing

## DATA AVAILABILITY

The genome sequences used in this study have all been previously deposited to NCBI SRA (IDs in the "SRA_IDs" tab of Data S1). Additionally, the exoenzyme SNPs and full sequences are available in Data S5. All paired phenotypic data for virulence assays and AMR are available in the "Phenotypes" tab of Data S1.

## ADDITIONAL FILES

The following material is available online.

### Supplemental Material

**Data S1 (mSystems01362-25-s0001.xlsx).** Accession ID and metadata.
**Data S2 (mSystems01362-25-s0002.xlsx).** TREE2 metrics.
**Data S3 (mSystems01362-25-s0003.txt).** IQ-TREE2 SNP tree with confidence supports.
**Data S4 (mSystems01362-25-s0004.xlsx).** Panaroo Scoary gene trait associations.
**Data S5 (mSystems01362-25-s0005.xlsx).** Exoenzyme sequences and SNPs.
**Data S6 (mSystems01362-25-s0006.xlsx).** Correlation analysis.
**Data S7 (mSystems01362-25-s0007.xlsx).** AMR mechanisms.
**Data S8 (mSystems01362-25-s0008.xlsx).** BWHPSA026 SNPs.

**Supplemental Figures (mSystems01362-25-s0009.pdf).** Figures S1-S5.
**Captions (mSystems01362-25-s0010.docx).** Captions for Data S1-S8.

## Open Peer Review

**PEER REVIEW HISTORY (review-history.pdf).** An accounting of the reviewer comments and feedback.

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
