## [Reviewer comments · mSystems]

Genomic comparison and phenotypic characterization of *Pseudomonas aeruginosa* isolates across environmental and diverse clinical isolation sites

Cristina Penaranda, Evan Brenner, Anne Clatworthy, Lisa Cosimi, Janani Ravi, and Deborah Hung

Corresponding Author(s): Cristina Penaranda, National Jewish Health

Review Timeline:

Submission Date:	September 23, 2025
Editorial Decision:	November 30, 2025
Revision Received:	December 18, 2025
Accepted:	January 14, 2026

Editor: Egon Ozer

Reviewer(s): The reviewers have opted to remain anonymous.

Transaction Report:

DOI: <https://doi.org/10.1128/msystems.01362-25>

Re: mSystems01362-25 (Mapping genetic and phenotypic diversity of *Pseudomonas aeruginosa* across clinical and environmental isolation sites)

Dear Dr. Cristina Penaranda:

The reviewers agreed on the quality and merit of the manuscript and offered focused suggestions for modifications to improve readability and clarity.

Revision Guidelines

Sincerely,
Egon Ozer
Editor
mSystems

Reviewer #1 (Comments for the Author):

The study "Mapping genetic and phenotypic diversity of *Pseudomonas aeruginosa* across clinical and environmental isolation sites" is an interesting study that provides valuable data; however, it requires some revisions.

1. In the title: The authors indicate they will perform "genetic mapping", which is inconsistent with their study. Knowing that

genetic mapping "is a diagram showing the relative location of genes and other genetic markers on a chromosome," and what the authors are doing is a genomic comparison. Therefore, I suggest changing "genetic mapping" to "genomic comparison."

2. Abstract. The authors describe the following paragraph as a summary: "In summary, our analyses of 125 diverse isolates suggest that the ability of *P. aeruginosa* to thrive across diverse niches is driven by broadly conserved genetic repertoire rather than niche-specific accessory genes". This summary does not accurately reflect your results. The authors should focus on the most relevant results and highlight them. On the other hand, we know because, as various studies have shown, *Pseudomonas* has a highly conserved core genome and a complex regulatory network that enables it to adapt to different environments.

3. Results: The authors say: "Strains did not cluster phylogenetically based on their anatomic site of infection (Figure 2A), in line with previous population-wide analysis". I believe the word "phylogenetically" is not appropriate when discussing clade formation based on the isolation site.

On the other hand, if the authors were to analyze the ST of the strains, they would probably realize that they are grouped by ST, as reported in the following article.

Gómez-Martínez et. al. 2023. Comparative Genomics of *Pseudomonas aeruginosa* Strains Isolated from Different Ecological Niches. *Antibiotics*. May 7;1, 866. 1- 21. Doi: 10.3390/antibiotics12050866

4. Discussion: The authors say: "Notably, our results do not align with previous data suggesting that host-adaptation results in genome reduction. It is important to note that the clinical data associated with the clinical strains in our collection does not differentiate between long-term versus short-term infections. Therefore, it remains possible that long-term adaptation to a mammalian host does result in genome reduction, as has been found in people with Cystic Fibrosis"

I believe this paragraph should be reworded, as the presence or absence of genes cannot determine whether strains will cause a short- or long-term infection, since, in an infectious process, one must consider the ecological triad "microorganism, host, and environment". Furthermore, we know that in cystic fibrosis, the strains' regulatory systems for alginate formation are involved, allowing *Pseudomonas* to persist for extended periods in these patients.

5. The authors say: "Classification of AMR by sequence analysis remains imperfect, and we must still rely on laboratory-based susceptibility testing, which is both costly and time-consuming since it must be done for each antibiotic individually".

The authors should be careful with this paragraph in the discussion and rewrite it, as they must keep in mind that genomic analysis only reveals the presence of genes, not whether they are expressed. Therefore, the phenotype determines whether those genes are expressed. These analyses have distinct value and provide different information, and they are complementary.

6. Conclusion: I suggest focusing the conclusion on its relevant results.

7. Suggest the authors upload their sequences to a database, as this would be a great contribution. And in the article, they can include the accession numbers for each strain.

Reviewer #2 (Comments for the Author):

This study investigates the genetic and phenotypic diversity of 125 *Pseudomonas aeruginosa* isolates from both clinical and environmental sources. The authors performed very elegant whole genome sequencing and phenotypic assays to assess virulence traits (motility, cytotoxicity, biofilm formation, pyocyanin production) and antimicrobial resistance (AMR) genes.

My main takeaways are:

1. Two major phylogenetic clades (Groups A and B) are described.
2. AMR was found exclusively in clinical isolates, not environmental ones, indicating antibiotic exposure as a driver of resistance.
3. The identification of a novel *exoU* allele

My only criticism is that the rationale for doing this was not well explained?

The study “Mapping genetic and phenotypic diversity of *Pseudomonas aeruginosa* across clinical and environmental isolation sites” is an interesting study that provides valuable data; however, it requires some revisions.

- 1. In the title:** The authors indicate they will perform “genetic mapping”, which is inconsistent with their study. Knowing that genetic mapping "is a diagram showing the relative location of genes and other genetic markers on a chromosome," and what the authors are doing is a genomic comparison. Therefore, I suggest changing "genetic mapping" to "genomic comparison."
- 2. Abstract.** The authors describe the following paragraph as a summary: “In summary, our analyses of 125 diverse isolates suggest that the ability of *P. aeruginosa* to thrive across diverse niches is driven by broadly conserved genetic repertoire rather than niche-specific accessory genes”. This summary does not accurately reflect your results. The authors should focus on the most relevant results and highlight them. On the other hand, we know because, as various studies have shown, *Pseudomonas* has a highly conserved core genome and a complex regulatory network that enables it to adapt to different environments.
- 3. Results:** The authors say: “Strains did not cluster phylogenetically based on their anatomic site of infection (Figure 2A), in line with previous population-wide analysis”. I believe the word "phylogenetically" is not appropriate when discussing clade formation based on the isolation site. On the other hand, if the authors were to analyze the ST of the strains, they would probably realize that they are grouped by ST, as reported in the following article.
Gómez-Martínez et. al. **2023**. Comparative Genomics of *Pseudomonas aeruginosa* Strains Isolated from Different Ecological Niches. *Antibiotics*. May 7;1, 866. 1- 21. Doi: 10.3390/antibiotics12050866
- 4. Discussion:** The authors say: “Notably, our results do not align with previous data suggesting that host-adaptation results in genome reduction. It is important to note that the clinical data associated with the clinical strains in our collection does not differentiate between long-term

versus short-term infections. Therefore, it remains possible that long-term adaptation to a mammalian host does result in genome reduction, as has been found in people with Cystic Fibrosis”

I believe this paragraph should be reworded, as the presence or absence of genes cannot determine whether strains will cause a short- or long-term infection, since, in an infectious process, one must consider the ecological triad "microorganism, host, and environment". Furthermore, we know that in cystic fibrosis, the strains' regulatory systems for alginate formation are involved, allowing *Pseudomonas* to persist for extended periods in these patients.

5. The authors say: “Classification of AMR by sequence analysis remains imperfect, and we must still rely on laboratory-based susceptibility testing, which is both costly and time-consuming since it must be done for each antibiotic individually”.

The authors should be careful with this paragraph in the discussion and rewrite it, as they must keep in mind that genomic analysis only reveals the presence of genes, not whether they are expressed. Therefore, the phenotype determines whether those genes are expressed. These analyses have distinct value and provide different information, and they are complementary.

6. **Conclusion:** I suggest focusing the conclusion on its relevant results.
7. **Suggest the authors upload their sequences to a database, as this would be a great contribution. And in the article, they can include the accession numbers for each strain.**

Response to reviewer comments,

We thank the reviewers for their insightful feedback, which we believe has led to meaningful improvements to the clarity of the manuscript. Below, we describe the changes made to address specific comments from the reviewers.

Reviewer #1 (Comments for the Author):

The study "Mapping genetic and phenotypic diversity of *Pseudomonas aeruginosa* across clinical and environmental isolation sites" is an interesting study that provides valuable data; however, it requires some revisions.

1. In the title: The authors indicate they will perform "genetic mapping", which is inconsistent with their study. Knowing that genetic mapping "is a diagram showing the relative location of genes and other genetic markers on a chromosome," and what the authors are doing is a genomic comparison. Therefore, I suggest changing "genetic mapping" to "genomic comparison."

We have updated the title to "Genomic comparison and phenotypic characterization of *Pseudomonas aeruginosa* isolates across environmental and diverse clinical isolation sites" to emphasize that we performed comparative genomics rather than genetic mapping.

2. Abstract. The authors describe the following paragraph as a summary: "In summary, our analyses of 125 diverse isolates suggest that the ability of *P. aeruginosa* to thrive across diverse niches is driven by broadly conserved genetic repertoire rather than niche-specific accessory genes". This summary does not accurately reflect your results. The authors should focus on the most relevant results and highlight them. On the other hand, we know because, as various studies have shown, *Pseudomonas* has a highly conserved core genome and a complex regulatory network that enables it to adapt to different environments.

We agree with the reviewers point. . While we do intend to highlight that conserved core genes are responsible for the versatility of *Pseudomonas aeruginosa*, this is not an entirely accurate summary of the overall study. We have rephrased the abstract to reflect our findings on the role of conserved vs. accessory gene requirement for adaptation, novel paired genome-phenotype resource, and the versatility of *P. aeruginosa* isolates in niche colonization, regardless of their origins.

3. Results: The authors say: "Strains did not cluster phylogenetically based on their anatomic site of infection (Figure 2A), in line with previous population-wide analysis". I believe the word "phylogenetically" is not appropriate when discussing clade formation based on the isolation site.

On the other hand, if the authors were to analyze the ST of the strains, they would probably realize that they are grouped by ST, as reported in the following article. Gómez-Martínez et. al. 2023. Comparative Genomics of *Pseudomonas aeruginosa* Strains Isolated from Different Ecological Niches. Antibiotics. May 7;1, 866. 1- 21. Doi: 10.3390/antibiotics12050866

The reviewer is correct that strains from the same ST will group together on the phylogenetic tree since STs are based on sequences of core genes. Our intention was to highlight that the strains did not cluster on the phylogenetic tree based on their isolation source, which might suggest that specific core gene sequences are

required for niche specialization. We have clarified this sentence to “Strains from the same anatomic site of infection did not cluster on the phylogenetic tree”.

4. Discussion: The authors say: "Notably, our results do not align with previous data suggesting that host-adaptation results in genome reduction. It is important to note that the clinical data associated with the clinical strains in our collection does not differentiate between long-term versus short-term infections. Therefore, it remains possible that long-term adaptation to a mammalian host does result in genome reduction, as has been found in people with Cystic Fibrosis"

I believe this paragraph should be reworded, as the presence or absence of genes cannot determine whether strains will cause a short- or long-term infection, since, in an infectious process, one must consider the ecological triad "microorganism, host, and environment". Furthermore, we know that in cystic fibrosis, the strains' regulatory systems for alginate formation are involved, allowing *Pseudomonas* to persist for extended periods in these patients.

We agree with this point. We have reworded the paragraph to clarify that, while our strains do not show significant differences in genome size or CDS content between environmental vs. clinical isolation sources, our strains are not stratified by acute vs. chronic infection, and we therefore cannot investigate the role of infection duration. While prior studies have shown genome reduction in long-term infections (e.g., people with CF), we cannot distinguish such effects with the limitations of our data.

5. The authors say: "Classification of AMR by sequence analysis remains imperfect, and we must still rely on laboratory-based susceptibility testing, which is both costly and time-consuming since it must be done for each antibiotic individually".

The authors should be careful with this paragraph in the discussion and rewrite it, as they must keep in mind that genomic analysis only reveals the presence of genes, not whether they are expressed. Therefore, the phenotype determines whether those genes are expressed. These analyses have distinct value and provide different information, and they are complementary.

We appreciate that gene expression does not necessarily follow gene presence, and indeed, the complex gene regulation of *P. aeruginosa* makes such inferences more difficult. We have revised the results and discussion to incorporate this nuance.

6. Conclusion: I suggest focusing the conclusion on its relevant results.

We thank the reviewer for this suggestion. We have refined the conclusion to focus on the major results and their interpretation as it pertains to the scientific questions we sought to address with this study.

7. Suggest the authors upload their sequences to a database, as this would be a great contribution. And in the article, they can include the accession numbers for each strain.

All sequencing data have already been uploaded to SRA. The per sample information can be found in Supplemental Data 1. We have now also included a “Data availability” section for clarity

Reviewer #2 (Comments for the Author):

This study investigates the genetic and phenotypic diversity of 125 *Pseudomonas aeruginosa* isolates from both clinical and environmental sources. The authors performed very elegant whole genome sequencing and phenotypic assays to assess virulence traits (motility, cytotoxicity, biofilm formation, pyocyanin production) and antimicrobial resistance (AMR) genes.

My main takeaways are:

- 1. Two major phylogenetic clades (Groups A and B) are described.**
- 2. AMR was found exclusively in clinical isolates, not environmental ones, indicating antibiotic exposure as a driver of resistance.**
- 3. The identification of a novel exoU allele**

My only criticism is that the rationale for doing this was not well explained?

We thank the reviewer for this comment. We have included additional text in the abstract, introduction, and discussion that we hope better reflects the goals and findings of our study.

Re: mSystems01362-25R1 (**Genomic comparison and phenotypic characterization of *Pseudomonas aeruginosa* isolates across environmental and diverse clinical isolation sites**)

Dear Dr. Cristina Penaranda:

Your manuscript has been accepted, and I am forwarding it to the ASM production staff for publication. Your paper will first be checked to make sure all elements meet the technical requirements. ASM staff will contact you if anything needs to be revised before copyediting and production can begin. Otherwise, you will be notified when your proofs are ready to be viewed.

Cover Image Submissions: If you would like to submit a potential Cover Image, please email a file and a short legend to mssystems@asmusa.org. Please note that we can only consider images that (i) the authors created or own and (ii) have not been previously published. By submitting, you agree that the image can be used under the same terms as the published article. Image File requirements: TIF/EPS, 7.5 inches wide by 8.25 inches tall (at least 2,250 pixels wide by 2,475 pixels tall), minimum 300 dpi resolution (600 dpi preferred), RGB, and no figure elements, e.g., arrows or panel labels. The legend should be a short description of the image, 1-2 sentences recommended. Please download and use this interactive template in Adobe to ensure that your proposed cover image meets our size requirements (<https://journals.asm.org/pb-assets/pdf-text-excel-files/ASM-Interactive-Sizing-Cover-Template-1715689791.pdf>).

Sincerely,
Egon Ozer
Editor
mSystems

Reviewer #1 (Comments for the Author):

Good job

Reviewer #2 (Comments for the Author):

My concerns are addressed. The data are needed.